# OpenReview forum: "InstructSafety: A Unified Framework for Building Multidimensional and Explainable Safety Detector through Instruction Tuning"
_EMNLP/2023/Conference — EMNLP 2023 Findings_

### Official Review · Reviewer_8L5c · 2023-08-02

**Soundness:** 3

**Excitement:**

4: Strong: This paper deepens the understanding of some phenomenon or lowers the barriers to an existing research direction.

**Paper Topic And Main Contributions:**

This paper focuses on the task of safety detection for natural language processing (NLP) models. The paper proposes a unified framework for 7 safety detection tasks based on the Flan-T5 model. The safety detection of the proposed method is mainly for societal safety such as bias and unfairness.

**Reasons To Accept:**

1. This paper design a unified framework for different safety evaluation task with seven tasks. The experiments are conducted in extensive datasets and the proposed method obtains state-of-the-art performance in those tasks. These results validate the effectiveness of the proposed method.
2. The proposed method for generating augmented data with APIs in Sec. 3.4 is reasonable and manifests how public APIs can be used to mitigate the bias.
3. The paper is well-written and easy-to-follow. The presentation is of high quality with good illustrations.

**Reasons To Reject:**

1. Although this paper provides a unified framework for different tasks. It is not clear about the ability of extension of the proposed framework when there is a new way of safety evaluation in the future. How such a framework can incorporate new task is still unknown.
2. Conceptually, this paper mainly focuses on the societal safety of NLP models, which is an aspect of NLP safety. It is not broad enough to claim that the proposed method is for safety detection with respect to other types of robustness such as adversarial robustness and so on.
3. The paper mentioned that "Note that we set a higher N for the subtask that has fewer instances" in L304 for the generated data. But why it is treated in this way is not mentioned. This is an important problem because the proposed method is basically a data augmentation method and the distribution of data sample is key to model performance. It would be interesting to how the data distribution affects the performance of the proposed method.
4. There is a lack of discussion on adversarial attacks in related work to differentiate the contribution of the proposed method. For example, adversarial attacks including hard-label,  soft-label, and white-box methods try to fool the victim model. They shall be included to make it clear for the general audience to grasp the highlighted societal safety aspect of this paper.

**Reproducibility:**

4: Could mostly reproduce the results, but there may be some variation because of sample variance or minor variations in their interpretation of the protocol or method.

**Reviewer Confidence:**

4: Quite sure. I tried to check the important points carefully. It's unlikely, though conceivable, that I missed something that should affect my ratings.

---

> ### Author Rebuttal · Authors · 2023-08-28
>
> We thank the reviewer for the detailed feedback and valuable suggestions to further improve our work.
>
> > Reject reason 1: Although this paper provides a unified framework for different tasks. It is not clear about the ability of extension of the proposed framework when there is a new way of safety evaluation in the future. How such a framework can incorporate new task is still unknown.
>
> Response: The incorporation of a new task into our framework is straightforward thanks to the convenience of instruction tuning. We only need to design some new instructions for the new task and collect some related data, while keeping the formats and data of other old tasks unchanged.
>
> > Reject reason 2 & Reject reason 4: Conceptually, this paper mainly focuses on the societal safety of NLP models, which is an aspect of NLP safety. It is not broad enough to claim that the proposed method is for safety detection with respect to other types of robustness such as adversarial robustness and so on. And there is a lack of discussion on adversarial attacks in related work to differentiate the contribution of the proposed method.
>
> Response: The term "safety detection" in our paper refers to the process of identifying safety issues within textual data from various sources. This implies that the input to safety detectors could originate from humans rather than models. In other words, our focus lies in detecting safety concerns within the text itself, rather than solely assessing the safety of the models. Consequently, the evaluation of the adversarial robustness of models may not fall within the scope of our definition. Moreover, our framework doesn't exclude the safety detection of adversarial text, although they are not the main focus of this paper. Based on the above reasons, we believe the discussion of adversarial attacks in related work is not necessary.
>
> Regrettably, we acknowledge our oversight in providing a precise elucidation of the term "safety detection" in our paper. We hereby commit to rectifying this and including a well-defined explanation of "safety detection" in our final paper.
>
> > Reject reason 3: The paper mentioned that "Note that we set a higher N for the subtask that has fewer instances" in L304 for the generated data. But why it is treated in this way is not mentioned. This is an important problem because the proposed method is basically a data augmentation method and the distribution of data sample is key to model performance. It would be interesting to how the data distribution affects the performance of the proposed method.
>
> Response: We have explained the reason in Line 299 - Line 303 of our paper. We set a higher N for the subtask that has fewer instances to harmonize the distribution of data across various subtasks, following previous work by Gupta et al. (2022). Our intention is for the trained model to exhibit a well-balanced performance across different subtasks, rather than being exclusively skewed towards the subtask with the most data.

---

### Official Review · Reviewer_XQAP · 2023-08-04

**Soundness:** 3

**Excitement:**

2: Mediocre: This paper makes marginal contributions (vs non-contemporaneous work), so I would rather not see it in the conference.

**Paper Topic And Main Contributions:**

The paper introduces INSTRUCTSAFETY, a unified framework that integrates seven common subtasks of safety detection based on different instructions. Additionally, the paper proposes an instruction collection comprising 39 datasets.

**Questions For The Authors:**

This paper introduces seven subtasks, but from my perspective, it seems there might be some overlap among Problem Detection, Problem Judgement, and Safety Comparison. Could you please clarify?

**Reasons To Accept:**

1. The paper applies instruction tuning to the issue of safety detection, which will support further integration with Language Models (LLMs) in the future.
2. The paper introduces an instruction collection with 39 datasets, which will facilitate future research.
3. The paper presents a simple and effective method for improving the robustness of safety detection models.

**Reasons To Reject:**

1. The paper claims that the proposed instruction collection consists entirely of high-quality data. However, diversity, another factor that impacts instruction tuning, is lacking in discussion and experimental analysis in this paper.
2. The experimental results are not sufficient. For the full-shot experiment, there are many datasets, but the performance comparison is only provided for one dataset per subtask, which calls the generalizability of the model into question.
3. The choice of Flan-T5 is not explained clearly. Would the usage of T5 produce similar results? Moreover, the experiments for the subtasks of Target Detection and Safety Explanation lack baseline results for comparison with Flan-T5.
4. It appears that the experiments were not conducted multiple times, and there is a lack of variance analysis for the results.

**Reproducibility:**

3: Could reproduce the results with some difficulty. The settings of parameters are underspecified or subjectively determined; the training/evaluation data are not widely available.

**Reviewer Confidence:**

4: Quite sure. I tried to check the important points carefully. It's unlikely, though conceivable, that I missed something that should affect my ratings.

---

> ### Author Rebuttal · Authors · 2023-08-28
>
> We thank the reviewer for the detailed feedback and valuable suggestions to further improve our work.
>
> > Reject reason 1: The paper claims that the proposed instruction collection consists entirely of high-quality data. However, diversity, another factor that impacts instruction tuning, is lacking in discussion and experimental analysis in this paper.
>
> Response: Qualitatively, we collect data from a total of 39 unique datasets, underscoring the inherent diversity present in our data. Quantitatively, we randomly sampled 10,000 instances from the training set and calculated their Distinct-4 value (the proportion of distinct 4-grams) to measure the diversity. The metric yielded a value of 0.66 when instructions are included and 0.87 when excluding the instructions we manually wrote for seven subtasks. This observation underscores the commendable level of diversity within our dataset. We will incorporate this result into our final paper.
>
> > Reject reason 2: The experimental results are not sufficient. For the full-shot experiment, there are many datasets, but the performance comparison is only provided for one dataset per subtask, which calls the generalizability of the model into question.
>
> Response: The datasets selected for performance comparison in our paper have been thoughtfully curated, ensuring the presence of appropriate baselines and metrics initially introduced in their respective original papers. While numerous datasets were employed for training purposes, it is worth noting that many of these lack comparable baselines and metrics as outlined in their original papers. This divergence substantially amplifies the complexity and resource requirements associated with our evaluation, particularly if we were to undertake the task of training and evaluating baselines for each of these distinct datasets.
>
> Moreover, as part of our efforts to validate the generalizability of our model, we have conducted zero-shot experiments, where our model exhibits superior performance compared to strong baselines including ChatGPT. Although we believe that the present results can already demonstrate the generalizability of our model to some extent, we will still try our best to include results on more datasets in our final paper according to your suggestion.
>
> > Reject reason 3: The choice of Flan-T5 is not explained clearly. Would the usage of T5 produce similar results? Moreover, the experiments for the subtasks of Target Detection and Safety Explanation lack baseline results for comparison with Flan-T5.
>
> Response: It is very likely the reviewer missed some results in our paper. We have conducted ablation study to explore the impact of selecting different base models. The result is shown in Table 11 within our paper, where we compare Flan-T5 with LLaMA, Alpaca and Vicuna and find that Flan-T5 achieves best performance. Our motivation to select Flan-T5 as our base model is that Flan-T5 has been trained on a variety of tasks with instructions, as stated in Line 231 - Line 237 in our paper. As T5 is not instruction-tuned, we naturally expect it to perform worse (the intuition has already been validated by comparing LLaMA and its instruction-tuned variants in Table 11).
>
> The supplementary results for T5 are presented in the subsequent table, accompanied by a corresponding presentation of results for Flan-T5, enabling straightforward comparison. Notably, each column within this table corresponds to the model's performance on a distinct subtask and the final column represents the average performance, mirroring the structure of Table 11 in our paper.
> | Model   | SJ   | PJ   | PD   | SC   | TD   | SD   | SE   | Avg. |
> | ------- | ---- | ---- | ---- | ---- | ---- | ---- | ---- | ---- |
> | T5-11B  | 95.9 | 83.0 | 70.3 | 77.0 | 79.5 | 71.4 | 42.2 | 74.2 |
> | T5-3B   | 94.9 | 71.8 | 69.6 | 75.7 | 77.7 | 69.4 | 41.9 | 71.6 |
> | T5-770M | 90.9 | 68.5 | 71.0 | 71.7 | 75.3 | 69.3 | 41.3 | 69.7 |
> | Flan-T5-11B  | 96.3 | 83.0 | 73.0 | 76.3 | 80.3 | 70.7 | 41.8 | 74.5 |
> | Flan-T5-3B   | 95.5 | 81.1 | 72.9 | 76.1 | 77.9 | 70.6 | 42.1 | 73.7 |
> | Flan-T5-770M | 93.4 | 78.6 | 72.1 | 73.3 | 76.7 | 70.3 | 41.8 | 72.3 |
>
> According to the table, it is evident that Flan-T5 outperforms T5 in all three scenarios, each with varying numbers of parameters. This outcome further substantiates our initial intuition that the fundamental ability to follow instructions confers significant benefits. We will incorporate the results into our final paper.
>
>
> The reason why we exclude the baseline results of the original Flan-T5 on the Target Detection and Safety Explanation task has been thoroughly expounded in Line 412 - Line 417 and Line 435 - Line 441 within our paper. The comparison with unsupervised baselines using token-level metrics such as BLEU would be naturally unfair on these two subtasks, given the presence of numerous credible answers for each sample in these two subtasks. Nonetheless, Safety-Flan-T5 is clearly much better than original Flan-T5 on safety detection according to the results on other subtasks. For the completeness, we put the results of the original Flan-T5 on the Target Detection and Safety Explanation task in the following tables.
>
> Results on the Target Detection task:
> | Model                | BLEU-2 | Rouge-L |
> | -------------------- | ------ | ------- |
> | Flan-T5-Large        | 49.0   | 48.7    |
> | Flan-T5-XL           | 28.5   | 29.3    |
> | Flan-T5-XXL          | 40.0   | 39.5    |
> | Safety-Flan-T5-Large | 76.7   | 73.9    |
> | Safety-Flan-T5-XL    | 77.9   | 75.4    |
> | Safety-Flan-T5-XXL   | 80.3   | 78.1    |
>
> We use 3-shot prompting to test the original Flan-T5 on the Target Detection task. The results indicate that Flan-T5 performs much worse than Safety-Flan-T5. The discernable advantage of Flan-T5-Large over Flan-T5-XXL arises from the fact that Flan-T5-Large's constrained capacity makes it considerably more prone to generating "None." as an output. The proportions of outputs that are "None." are 67%, 34% and 51% for Flan-T5-large, Flan-T5-XL and Flan-T5-XXL respectively. Meanwhile, "None." accounts for a substantial 59% of the ground truth labels, thereby contributing to the superior performance of Flan-T5-Large.
>
> Results on the Safety Explanation task:
> | Model                | BLEU-4 | F1    |
> | -------------------- | ------ | ----- |
> | Flan-T5-Large        | 2.28   | 22.93 |
> | Flan-T5-XL           | 3.20   | 27.18 |
> | Flan-T5-XXL          | 2.76   | 24.87 |
> | Safety-Flan-T5-Large | 15.08  | 41.79 |
> | Safety-Flan-T5-XL    | 15.47  | 42.14 |
> | Safety-Flan-T5-XXL   | 15.34  | 41.83 |
>
> We use 3-shot prompting to test the original Flan-T5 on the Safety Explanation task. Unsurprisingly, Safety-Flan-T5 outperforms Flan-T5 by a large margin. The reason Flan-T5-XL performs better than Flan-T5-Large and Flan-T5-XXL may be that the distribution of explanations generated by Flan-T5-XL happens to be closer to the distribution of ground truth explanations, which can also explain why Safety-Flan-T5-XL performs slightly better than Safety-Flan-T5-XXL on this dataset.
>
>
>
> > Reject reason 4: It appears that the experiments were not conducted multiple times, and there is a lack of variance analysis for the results.
>
> Response: We have conducted the experiments multiple times for Safety-Flan-T5. We set three different seeds for training Safety-Flan-T5 of different sizes. The result is shown in the subsequent table. Notably, each column within this table corresponds to the model's performance on a distinct subtask and the final column represents the average performance, mirroring the structure of Table 11 in our paper.
> | Model               | SJ           | PJ           | PD           | SC           | TD           | SD           | SE           | Avg.         |
> | ------------------- | ------------ | ------------ | ------------ | ------------ | ------------ | ------------ | ------------ | ------------ |
> | Safety-Flan-T5-11B  | 96.4$\pm$0.1 | 83.4$\pm$0.6 | 72.3$\pm$0.6 | 76.9$\pm$0.4 | 80.0$\pm$0.4 | 70.0$\pm$0.5 | 41.8$\pm$0.1 | 74.4$\pm$0.1 |
> | Safety-Flan-T5-3B   | 95.1$\pm$0.3 | 81.2$\pm$0.2 | 72.1$\pm$0.9 | 76.2$\pm$0.1 | 78.0$\pm$0.2 | 70.4$\pm$0.5 | 42.1$\pm$0.0 | 73.6$\pm$0.2 |
> | Safety-Flan-T5-770M | 93.5$\pm$0.2 | 79.2$\pm$0.8 | 72.4$\pm$0.4 | 73.5$\pm$0.3 | 76.9$\pm$0.2 | 70.5$\pm$0.2 | 41.8$\pm$0.0 | 72.5$\pm$0.2 |
>
> Based on the results provided within this table, it is evident that the performance of Safety-Flan-T5 remains consistently stable, exhibiting minimal variances across the majority of subtasks. Notably, the standard variation in average performance does not exceed 0.2 across varying sizes of  Safety-Flan-T5. We will incorporate the results into our final paper. Note that we don't need to conduct experiments for unsupervised baselines (e.g., ChatGPT) compared in each subtask multiple times because we have set the temperature to 0.
>
>
> Question: This paper introduces seven subtasks, but from my perspective, it seems there might be some overlap among Problem Detection, Problem Judgement, and Safety Comparison. Could you please clarify?
>
> Response: The Problem Detection task aims to detect and list possible safety problems of the input while the Problem Judgement task aims to judge whether the input text exhibits some specific safety problem. Therefore the Problem Detection task can be viewed as an integration of multiple different Problem Judgement tasks. The Safety Comparison task aims to compare the safety of multiple input texts (e.g., two toxic comments with different degrees of toxicity), which is clearly different from the Problem Detection and Problem Judgement tasks.

---

### Official Review · Reviewer_UwgW · 2023-08-05

**Soundness:** 3

**Excitement:**

4: Strong: This paper deepens the understanding of some phenomenon or lowers the barriers to an existing research direction.

**Paper Topic And Main Contributions:**

This paper introduces INSTRUCTSAFETY, a unified framework that integrates 7 common subtasks of safety detection based on different instructions. Besides, it proposes a simple but effective method to augment adversarial samples that can improve the model’s robustness. The authors conducted extensive experiments to validate the proposed method.

**Questions For The Authors:**

Question A:
What is the performance of the generated adversarial examples?

Question B:

Are there other ways to make the method proposed in this paper more effective in removing bias?

Question C:
Are the decisions of the fine-tuned Safety-Flan-T5XL reliable?

**Reasons To Accept:**

1.	The research topic is interesting.

2.	The experimental results are promising.

**Reasons To Reject:**

1.	The performance of adversarial examples was not evaluated.
As one of the main contributions of the author, the robustness of the model is enhanced by generating adversarial examples. However, the performance of adversarial examples has not been evaluated. What is the performance of the generated adversarial examples?

2.	The data set generation method needs further validation.
The authors use the fine-tuned Safety-Flan-T5XL to identify any generated descriptions that are falsely predicted as unsafe, and incorporate these descriptions into their training set for the safety judgement task. However, the authors did not examine whether relying on the decision of the fine-tuned Safety-Flan-T5XL is reliable.

**Reproducibility:**

3: Could reproduce the results with some difficulty. The settings of parameters are underspecified or subjectively determined; the training/evaluation data are not widely available.

**Reviewer Confidence:**

3: Pretty sure, but there's a chance I missed something. Although I have a good feel for this area in general, I did not carefully check the paper's details, e.g., the math, experimental design, or novelty.

---

> ### Author Rebuttal · Authors · 2023-08-28
>
> We thank the reviewer for the detailed feedback and valuable suggestions to further improve our work.
>
> > Reject reason 1 & Question A: The performance of adversarial examples has not been evaluated. What is the performance of the generated adversarial examples?
>
> Response: It is very likely the reviewer missed some results in our paper. We present the performance of the generated adversarial examples within our ablation study in Table 10. The results depicted in Table 10 underscore the noteworthy advantages conferred by the generated adversarial examples on the model's overall robustness. For example, the  accuracy of Safety-Flan-T5-Large when dealing with non-hateful cases on the HateCheck dataset has been raised from 81.0 to 84.3 with the integration of  the generated adversarial examples.
>
> > Reject reason 2 & Question C: The data set generation method needs further validation. The authors use the fine-tuned Safety-Flan-T5XL to identify any generated descriptions that are falsely predicted as unsafe, and incorporate these descriptions into their training set for the safety judgement task. However, the authors did not examine whether relying on the decision of the fine-tuned Safety-Flan-T5XL is reliable.
>
> Response: The reviewer might misunderstand our method. We suppose all descriptions generated by gpt-3.5-turbo and text-davinci-003 are actually safe. Based on this premise, the descriptions flagged as unsafe by Safety-Flan-T5XL are all falsely predicted samples, which are adversarial samples that are beneficial to improve Safety-Flan-T5's robustness. Therefore, it is meaningless and impossible to examine the reliability of Safety-Flan-T5XL's decision.
>
> To quantitatively support our hypothesis that all descriptions generated by gpt-3.5-turbo and text-davinci-003 are actually safe, we randonly sampled 50 descriptions with fixed patterns (25 from gpt-3.5-turbo and 25 from text-davinci-003) and 50 free-form descriptions  (25 from gpt-3.5-turbo and 25 from text-davinci-003) that were predicted as unsafe by Safety-Flan-T5XL. We manually checked whether these descriptions are actually safe and found that only 1 out of the total 100 descriptions could be considered unsafe. The result signifies that 99% of the unsafe descriptions flagged by Safety-Flan-T5XL are actually safe, which strongly supports our hypothesis.
>
>
> > Question B: Are there other ways to make the method proposed in this paper more effective in removing bias?
>
> Response: Constructing adversarial examples is a very convenient and effective method to help remove bias in safety detectors. While other methods, such as manipulating model predictions based on a fairness constraint (Pleiss et al., "On Fairness and Calibration"), and introducing additional restrictions during training (Zhao et al., "Men Also Like Shopping: Reducing Gender Bias Amplification using Corpus-level Constraints"), are either hard to apply, or can only remove bias in limited aspects. Therefore, we resort to large language models for constructing adversarial examples, a strategy that proves both efficacious and straightforward to employ.

---

### Meta-Review · Area_Chair_72oP · 2023-09-15

**Recommendation:** 4

**Metareview:**

This paper presents a safety detection framework, InstructSafety, comprising 7 common subtasks in safety detection, as well as a pre-processed dataset for instruction tuning and a fine-tuned model (Safety-Flan-T5). Reviewers agree that these contributions are useful and relevant, and that the paper is well-written and interesting. All three reviewers agree that the approach is generally sound, while two reviewers are highly excited about this work (score of 4).

The main criticisms of the work relate to the evaluation of the fine-tuned model. Reviewer UwgW initially had questions about the effect of adversarial examples and the data set generation method, although these issues appear to be resolved by the author response. Reviewer XQAP asks about extending the evaluation to more than one dataset per subtask, comparing Flan-T5 with other models, and about the variance within experimental results. The latter two issues are clearly and quantitatively addressed in the author rebuttal. The authors are encouraged to address the first point (i.e., to explain why certain datasets were included and not others, or to consider additional experiments) in their revision. Reviewer 8L5c also highlights the need to carefully define the scope of “safety detection” and to explain why the present definition excludes model robustness against adversarial attacks, etc. The authors are also encouraged to explain how additional tasks can be added to the framework, and to report the diversity of the data included in the instruction-tuning dataset.

Pros
- Unified framework comprising 7 tasks for safety detection
- Release of pre-processed dataset containing 39 human-annotated sub-datasets for instruction tuning
- Safety-Flan-T5 shows state of the art performance

Cons
- Evaluation is somewhat limited in that it includes only one dataset per subtask
- Scope of the “safety detection” problem needs clarification
- Further discussion needed around how to incorporate new or other tasks (e.g. adversarial attacks)

---

### Decision · Program_Chairs · 2023-10-07

**Decision:**

Accept-Findings

**Comment:**

This paper presents a safety detection framework, InstructSafety, comprising 7 common subtasks in safety detection, as well as a pre-processed dataset for instruction tuning and a fine-tuned model (Safety-Flan-T5). Reviewers agree that these contributions are useful and relevant, and that the paper is well-written and interesting. All three reviewers agree that the approach is generally sound, while two reviewers are highly excited about this work (score of 4).

The main criticisms of the work relate to the evaluation of the fine-tuned model. Reviewer UwgW initially had questions about the effect of adversarial examples and the data set generation method, although these issues appear to be resolved by the author response. Reviewer XQAP asks about extending the evaluation to more than one dataset per subtask, comparing Flan-T5 with other models, and about the variance within experimental results. The latter two issues are clearly and quantitatively addressed in the author rebuttal. The authors are encouraged to address the first point (i.e., to explain why certain datasets were included and not others, or to consider additional experiments) in their revision. Reviewer 8L5c also highlights the need to carefully define the scope of “safety detection” and to explain why the present definition excludes model robustness against adversarial attacks, etc. The authors are also encouraged to explain how additional tasks can be added to the framework, and to report the diversity of the data included in the instruction-tuning dataset.

Pros
- Unified framework comprising 7 tasks for safety detection
- Release of pre-processed dataset containing 39 human-annotated sub-datasets for instruction tuning
- Safety-Flan-T5 shows state of the art performance

Cons
- Evaluation is somewhat limited in that it includes only one dataset per subtask
- Scope of the “safety detection” problem needs clarification
- Further discussion needed around how to incorporate new or other tasks (e.g. adversarial attacks)